



# Extended enthalpy formulations in the ice flow model ISSM version 4.17: discontinuous conductivity and anisotropic SUPG

Martin Rückamp[1], Angelika Humbert[1,2], Thomas Kleiner[1], Mathieu Morlighem[3], and Helene Seroussi[4]

[1]Alfred-Wegener-Institut Helmholtz-Zentrum für Polar- und Meeresforschung, Bremerhaven, Germany
[2]University of Bremen, Bremen, Germany
[3]Department of Earth System Science, University of California, Irvine, California, USA
[4]Jet Propulsion Laboratory, California Institute of Technology, Pasadena, California, USA

**Correspondence:** Martin Rückamp (martin.rueckamp@awi.de)

**Abstract.** The thermal state of an ice sheet is an important control on its past and future evolution. Some parts of the ice sheets may be polythermal, leading to discontinuous properties at the cold–temperate transition surface (CTS). These discontinuities require a careful treatment in ice sheet models (ISMs). Additionally, the highly anisotropic geometry of the 3D elements in ice sheet modelling poses a problem for stabilization approaches in advection dominated problems. Here, we present ex-

5 tended enthalpy formulations within the finite-element Ice Sheet System Model (ISSM) that show a better performance to earlier implementations. In a first polythermal-slab experiment, we found that the treatment of the discontinuous conductivities at the CTS with a geometric mean produce more accurate results compared to the arithmetic or harmonic mean. This improvement is particularly efficient when applied to coarse vertical resolutions. In a second ice dome experiment, we find that the numerical solution is sensitive to the choice of stabilization parameters in the well-established Streamline Upwind

10 Petrov–Galerkin (SUPG) method. As standard literature values for the SUPG stabilization parameter are not accounting for the highly anisotropic geometry of the 3D elements in ice sheet modelling, we propose a novel Anisotropic SUPG (ASUPG) formulation. This formulation circumvents the problem of high aspect-ratio by treating the horizontal and vertical directions separately in the stabilization coefficients. The ASUPG method provides accurate results for the thermodynamic equation on geometries with very small aspect ratios like ice sheets.

## 1 Introduction

Ice sheets and glaciers are important components of the climate system. Their evolution is one of the primary sources of sea-level change (Church et al., 2013). Besides the interactions of the ice sheet with the environment, changes in ice flow can alter the internal thermal state of the ice, which in turn can affect ice dynamics. Therefore thermo-mechanical numerical modelling

20 of ice sheets is a crucial tool to understand both their past and future evolution.





Ice sheets and glaciers can exhibit a polythermal state that includes both cold (below the pressure melting point) and temperate (at the pressure melting point) domains, separated by the cold–temperate transition surface (CTS) (Blatter and Hutter, 1991). In temperate ice, the heat generated by viscous deformation leads to a change of phase (Fowler, 1984; Blatter and Hutter, 1991), hence temperate ice contains liquid water. The decrease of the ice viscosity with increasing content of liquid water in temperate ice in turn enhances ice flow (Duval, 1977), especially if the temperate ice is present in basal layers, where shear deformation is largest.

Modern state-of-the-art ice sheet models (ISMs) simulate the thermal state according to the enthalpy method originally formulated in Aschwanden et al. (2012) and further developed and verified in Kleiner et al. (2015), Blatter and Greve (2015), Greve and Blatter (2016) and Hewitt and Schoof (2017). The main advantage of this formulation is the elimination of tracking the CTS, as both cold and temperate ice domains are handled within one equation for the enthalpy $E$; temperature $T$ and liquid water fraction $\omega$ are diagnostically computed from enthalpy. An increasing number of ice flow models is adopting an enthalpy scheme (e.g. Aschwanden et al., 2012; Brinkerhoff and Johnson, 2013; Seroussi et al., 2013; Kleiner et al., 2015; Greve and Blatter, 2016; Hoffman et al., 2018).

In ISMs, the governing thermodynamic equation are discretized, e.g. using the finite element method (FEM). Special care has to be taken when employing the FEM to the parabolic thermodynamic equation. Numerical instabilities inherent to the advection component of this equation tend to occur without stabilizing the standard Galerkin finite element method. To maintain stabilization, the popular Streamline Upwind Petrov–Galerkin (SUPG) method is often employed to the discrete problem (Brooks and Hughes, 1982). Although the SUPG method is well-established for convection-dominated problems, the optimal parameter choices are still subject of extensive research (e.g. Tezduyar and Osawa, 2000; John and Knobloch, 2007). The aspect ratio of anisotropic grid-cells in the FEM is particularly problematic, and error analysis is often restricted to two dimensions (e.g. John et al., 2018). Moreover, current mathematical and numerical analyses are not always general enough to apply to real-world applications (John et al., 2018).

ISMs are dealing with very thin geometries, since the ice vertical extent (up to ∼4 km) is much smaller than its lateral extent (up to several thousands of kilometres). As a consequence, 3D elements are frequently highly anisotropic and pose a challenging problem in order to maintain stabilization. For instance, a non-optimal choice of stabilization parameters could result either in under- or over-stabilization of the numerical solution. As a consequence of increasing computer power and modern models frequently relying on the FEM, Helanow and Ahlkrona (2018) investigated the accuracy and robustness of linear equal order finite elements discretization with Galerkin least-squares (GLS) stabilization on the Stokes equation system with anisotropic meshes. They found that common literature values for this stabilization scheme perform well on simple domains. However, on more complex geometries, in particular, at the ice margin of outlet glaciers, the choice of standard parameters results in significant oscillations in the vertical component of the surface velocity.

Beside the need for efficient stabilization in FEM, the phase change in the enthalpy formation leads to discontinuous thermal properties. This feature needs to be handled with care when seeking a numerical solution. Of particular concern are discontinuities of the thermal conductivity (Patankar, 1980; Voller and Swaminathan, 1993; Voller, 2001; Nield and Bejan, 2013). Kleiner et al. (2015) mentioned, that treating the discontinuous conductivity at the CTS as an arithmetic mean causes non-plausible





oscillations in the enthalpy solution that are visible, e.g. in a time-varying CTS position. Our work is indeed inspired by the current lack of accuracy of the simulated vertical enthalpy profile to the analytical solution obtained with the ice flow model ISSM with a coarse vertical resolution ($\Delta z$=10 m, Kleiner et al., 2015, see Fig. 4 (upper row) therein).

We describe and analyze here recent developments designed to obtain an enthalpy formulation within the finite-element model ISSM (Ice Sheet System Model, Larour et al., 2012) that performs well over a wide range of grid aspect ratios in advection dominated problems. The focus of this work is twofold: on the one hand, we focus on treatments of discontinuous conductivities at the CTS. Here, we test three formulations for the discontinuous conductivity proposed in Nield and Bejan (2013) for porous medium. On the other hand, we test SUPG formulations on thin geometries like ice sheets. Therefore, we run sensitivity experiments to test distinct parameter choices. One component of this study is the presentation of a novel anisotropic

SUPG (ASUPG) method in ice sheet modelling that decouples the vertical from the horizontal direction to account for their different scales. The formulations presented are extensions of the current implementations within the ice flow model ISSM (version 4.17) compared to Seroussi et al. (2013) and Kleiner et al. (2015).

## 2 Theory and Background

### 2.1 Mathematical model

Let $\Omega(t) \subseteq \mathbb{R}^3$ be a three-dimensional domain with $t \in [0, t_{\max}]$. The equations are given in Cartesian coordinates, in which $x$ and $y$ are in the horizontal plane, and $z$ is positive upward. The enthalpy balance equation reads

$$\varrho_i \left( \frac{\partial E}{\partial t} + \boldsymbol{v} \cdot \nabla E \right) = -\nabla \cdot \boldsymbol{q}_i + \Psi, \tag{1}$$

with the specific enthalpy (internal energy) $E$, the ice velocity vector $\boldsymbol{v} = (v_x, v_y, v_z)$, the ice density $\varrho_i$, the conductive flux $\boldsymbol{q}_i$, and the heat source by internal deformation $\Psi$. The enthalpy field equation of the ice–water mixture depends on whether

the mixture is cold or temperate. The conductive flux in cold ice is represented by Fourier's law but replacing temperature $T$ by $E$. In the temperate domain, the conductive flux is the latent heat flux (for simplicity we ignore here the sensible heat flux caused by variations in the pressure melting point temperature $T_{\mathrm{pmp}}(p)$):

$$\boldsymbol{q}_i = -K_{\mathrm{eff}} \nabla E = - \begin{cases} K_c \nabla E & E < E_{\mathrm{pmp}} \\ K_0 \nabla E & E \geq E_{\mathrm{pmp}} \end{cases}, \tag{2}$$

where $E_{\mathrm{pmp}}$ is the specific enthalpy at the pressure melting point and $K_c = k_i/c_i$ the enthalpy conductivity in cold ice, where

$k_i$ is the temperature conductivity and $c_i$ the specific heat capacity. The temperature dependence of the heat conductivity and specific heat capacity is neglected. The temperate ice conductivity $K_0$ remains poorly constrained as laboratory experiments and field observations are scarce. In this study, $K_0$ is simply varied to test its sensitivity on the polythermal structure.

At the upper surface, Dirichlet boundary conditions are imposed. The type of basal boundary condition (Neumann or Dirichlet) is time dependent and follows the decision chart for local basal conditions given in Aschwanden et al. (2012). However,

the boundary conditions for the conducted experiments in this study are specified below.





## 2.2 Finite element formulation

In ISSM (Larour et al., 2012; Seroussi et al., 2013), the enthalpy equation (Eq. 1) is discretized with piecewise bilinear P1×P1 elements and stabilized using the SUPG method according to Franca et al. (2006). The stabilized finite element methods for Eq. 1 can be written as: find $E \in H_0^1(\Omega)$ such that

$$B(E, w) = F(w) \quad \forall w \in H_0^1(\Omega), \tag{3}$$

where

$$B(E, w) = \left(\frac{\partial E}{\partial t} + \boldsymbol{v} \cdot \nabla E, w\right) + K_{\text{eff}}(\nabla E, \nabla w) + S(E, w), \tag{4}$$

$$F(w) = (\Psi, w), \tag{5}$$

where $(\cdot, \cdot)$ is the inner product of the Hilbert space $H_0^1(\Omega)$ of square integrable functions and derivatives, and are zero on the domain boundary. The term $S(E, w)$ is added to the standard variational formulation such that consistency is preserved and numerical stability enhanced. There are different stabilization schemes that are usually considered (Franca et al., 2006); here we rely on the SUPG method:

$$S^{\text{SUPG}}(E, w) = \sum_K \tau_K(-K_{\text{eff}}\Delta E + \boldsymbol{v} \cdot \nabla E - \Psi, \boldsymbol{v} \cdot \nabla w)_K \tag{6}$$

where $K$ denotes an arbitrary element of the triangulation $T_h$, $\tau_K$ is a stability coefficient and $(\cdot, \cdot)_K$ denotes integration over $K$. Please note, that for bilinear elements $\Delta E = 0$.

The stabilization parameter, $\tau_K$ is formulated as follows (Brooks and Hughes, 1982; Franca et al., 2006)

$$\tau_K = \frac{h_K}{2|\boldsymbol{v}|}\xi(\text{Pe}_K), \tag{7}$$

$$\text{Pe}_K = \frac{m_k|\boldsymbol{v}|h_K}{2K_{\text{eff}}}, \tag{8}$$

$$\xi(\text{Pe}_K) = \begin{cases} \text{Pe}_K & 0 \le \text{Pe}_K < 1 \\ 1 & \text{Pe}_K \ge 1 \end{cases}. \tag{9}$$

$h_K$ is a characteristic dimension of element $K$ (referred to as local mesh parameter), $\xi$ is an upwind function and $\text{Pe}_K$ is the local Peclet number. The usual Peclet definition is modified by including $m_k$, which takes into account the effect of the degree of interpolation, $k$. For linear interpolations, $m_{k=1}$ is $1/3$ (Franca et al., 1992). For the velocity norm $|\boldsymbol{v}|$ we use the euclidean norm.

## 2.3 Anisotropic SUPG

The standard stabilization techniques were initially developed for isotropic meshes, which essentially require that the elements have a similar size in all spatial directions. Once the elements become anisotropic or distorted, the local mesh parameter





plays an important role in the calculation of stabilizing coefficients. Various definitions have been utilized based on e.g. the maximum edge length, minimum edge length, circumradius of an element, and the element length aligned with the upwind direction (e.g., Mittal, 2000; Knobloch, 2008; Brinkerhoff and Johnson, 2015). Apart from that, Becker and Rannacher (1995)

and Blasco (2008) introduced stabilization coefficients for GLS diffusion that cover geometrical information from different spatial directions. These definitions do not cover the element characteristic that stems from thin 3D elements. In ice sheet modelling, 3D meshes are generally formed by extruding vertically triangular meshes, leading to prismatic elements that are highly anisotropic since the vertical extent is typically one or two orders of magnitude smaller than the horizontal extent. Typically, 15 to 20 horizontal layers are used, with thinner layers close to the base. Considering a one-kilometer thick ice sheet,

that is discretized in the horizontal direction between $0.5\,\mathrm{km}$ and $20\,\mathrm{km}$, aspect ratios could exceed 100. Taking the maximum edge length as the local mesh parameter $h_K$, which is a default choice for isotropic elements, would lead to over-stabilization, while taking the minimum edge length as $h_K$ would result in under-stabilization.

In order to develop a new SUPG stabilized method for anisotropic meshes, which accounts for geometrical information from the mesh, we consider a Cartesian three-dimensional mesh with prismatic elements. In doing so, we split the traditional SUPG

formulation into a horizontal and vertical direction with the stabilization parameters $\tau_K^{\mathrm{horizontal}}$ and $\tau_K^{\mathrm{vertical}}$, respectively. Relying on the ideas for stabilization parameters in different spatial direction by Becker and Rannacher (1995) and Blasco (2008), the anisotropic SUPG (ASUPG) stabilization term $S(E, w)$ is written as

$$S^{\mathrm{ASUPG}}(E, w) = S_1(E, w) + S_2(E, w), \tag{10}$$

where

$$S_1(E, w) = \sum_K \left( \begin{pmatrix} v_x\sqrt{\tau_K^{\mathrm{horizontal}}} \\ v_y\sqrt{\tau_K^{\mathrm{horizontal}}} \\ v_z\sqrt{\tau_K^{\mathrm{vertical}}} \end{pmatrix} \cdot \nabla E, \begin{pmatrix} v_x\sqrt{\tau_K^{\mathrm{horizontal}}} \\ v_y\sqrt{\tau_K^{\mathrm{horizontal}}} \\ v_z\sqrt{\tau_K^{\mathrm{vertical}}} \end{pmatrix} \cdot \nabla w \right)_K, \tag{11}$$

$$S_2(E, w) = \sum_K \left( -K_{\mathrm{eff}}\Delta E - \Psi, \begin{pmatrix} v_x\tau_K^{\mathrm{horizontal}} \\ v_y\tau_K^{\mathrm{horizontal}} \\ v_z\tau_K^{\mathrm{vertical}} \end{pmatrix} \cdot \nabla w \right)_K. \tag{12}$$

The stabilization parameters $\tau_K^{\mathrm{horizontal}}$ and $\tau_K^{\mathrm{vertical}}$ are similar to those calculated in Eqs. 7, 8, and 9, but the ASUPG approach replaces the local mesh parameter $h_K$ with the characteristic horizontal and vertical dimension of the element $K$. That means $h_k$ is replaced by $h_K^{\mathrm{horizontal}}$ and $h_K^{\mathrm{vertical}}$ in the two spatial directions. Here, both are calculated as the maximum extent of the

element $K$ in the respective directions.

### 2.4 Treatment of discontinuous conductivity

Since the conductivity is discontinuous at the CTS, special attention must be paid to the treatment of the effective conductivity $K_{\mathrm{eff}}$ in Eq. 2. The effective thermal conductivity of the solid-fluid system is related to the conductivity of the solid (ice), $K_c$,





and to the conductivity of the fluid (water), $K_0$, and depends in a complex way on the geometry of the medium. In Nield and
Bejan (2013), three models are proposed:

1. The effective thermal conductivity is the weighted arithmetic mean:

$$K_{\text{eff}}^{\text{arithmetic}} = \theta K_0 + (1-\theta)K_c. \tag{13}$$

2. The effective thermal conductivity is the weighted harmonic mean:

$$\frac{1}{K_{\text{eff}}^{\text{harmonic}}} = \frac{\theta}{K_0} + \frac{(1-\theta)}{K_c}. \tag{14}$$

3. The effective thermal conductivity is given by the weighted geometric mean:

$$K_{\text{eff}}^{\text{geometric}} = K_0^{\theta} K_c^{(1-\theta)}. \tag{15}$$

The weighting term $\theta$ indicates the volume occupied by liquid water. The discontinuous conductivity model is only evaluated
for elements that contain a CTS.

The applicability of the three models is controversial in the literature and depends strongly on the problem. However, Nield
and Bejan (2013) recommend the arithmetic mean if the heat conduction in the solid and fluid phases occurs in parallel. On
the other hand, the harmonic mean is appropriate if the structure and orientation of the porous medium is such that the heat
conduction takes place in series, with all of the heat flux passing through both solid and fluid. Since heat conduction through
porous media is likely a combination of both structures, a geometric mean can be interpreted as accounting for both processes as
it always results in a value in between an arithmetic and harmonic mean (assuming $K_c \neq K_0$). When $K_c$ and $K_0$ are equal, the
three models give the same effective thermal conductivity. For the limit case, where $K_0 \to 0$, the harmonic and geometric mean
imply insulating properties as $K_{\text{eff}} \to 0$ and no heat flux occurs across the interface; the arithmetic mean retains a non-zero
flux.

## 3 Experiments

We run several experiments with the emphasis to test our modifications in ISSM on accuracy and on stability. The discontinuous
conductivity treatments are verified against an analytical solution within a polythermal slab experiment. As this experiment
results effectively in a one-dimensional vertical experiment, it is not suitable to test the SUPG parameter choices. Therefore,
we setup a synthetic second ice dome experiment with variations in the topography. Constants and model parameters used in
the experiments are summarized in Tab. 1.

### 3.1 Polythermal slab

We repeat the well-established polythermal sided slab experiment proposed in Greve and Blatter (2009) and already applied to
ISSM in Kleiner et al. (2015). The setup poses a reasonable situation in glacier modelling with an intra-glacial CTS. The model





domain consists of a 200 m thick and $4°$ downward inclined ice slab. The horizontal velocity $v_x$ is prescribed as an analytical expression (from $5\,\mathrm{m\,a^{-1}}$ at the base towards $\sim 38\,\mathrm{m\,a^{-1}}$ at the surface, while $v_y = 0\,\mathrm{m\,a^{-1}}$), and the vertical velocity is set to be constant and equal to $v_z = -0.2\,\mathrm{m\,a^{-1}}$. In addition, the geothermal heat flux is set to be zero during the model run so that
the englacial strain heating is the only source of energy in the enthalpy balance equation.

An analytical solution for the steady-state enthalpy profile based on the solution of Greve and Blatter (2009) leads to a CTS position $18.95\,\mathrm{m}$ above the bed. In our experiments, the conductivity ratio $K_0/K_c$ is varied from $10^{-1}$ to $10^{-5}$. The simulations are performed on equidistant horizontal layers using different vertical resolutions $\Delta z = (10, 5, 2, 0.5)\,\mathrm{m}$. In this set-up, no stabilization is applied, i.e. the term $S(E, w)$ in Eq. 4 is ignored. Please note that the analytical solution considers
$K_0 = 0\,\mathrm{kg\,m^{-1}\,s^{-1}}$. In this experiment, we apply a thermal steady-state solver (i.e. $\partial/\partial t = 0$ in Eq. 1). Comparisons of results when applying a transient solver or a steady-state solver revealed no difference in the steady-state enthalpy profile.

## 3.2 Ice dome

In this experiment, a more realistic set-up than the polythermal slab experiment is considered with a three-dimensional ice dome based on the Vialov profile (Vialov, 1958). Other settings and parameters are borrowed from the EISMINT Phase 2
benchmark (Payne et al., 2000). The surface $z_s$ and bedrock $z_b$ of the entire ice sheet are defined as:

$$z_b(x,y) = 0, \tag{16}$$

$$z_s(x,y) = h(x,y)$$
$$= 10 + h_{\max}\left(1 - (r/r_{\max})^{(n+1)/n}\right)^{n/(2n+2)}, \tag{17}$$

with the ice thickness $h(x,y)$, the maximum ice thickness $h_{\max}$, the radius $r = \sqrt{x^2 + y^2}$, the maximum extent $r_{\max}$, and the
Glen exponent $n$. The summit of the ice dome is located at $(x,y) = (0,0)$.

In this experiment, a thermo-mechanical coupling is considered. The Glen–Steinemann power-law rheology (Steinemann, 1954; Glen, 1955) is used for the deformation of ice. The ice viscosity reads

$$\eta = \frac{1}{2}A^{-1/n}\dot{\varepsilon}_{\mathrm{eff}}^{-2/n}, \tag{18}$$

where $A$ is the flow rate factor and $\dot{\varepsilon}_{\mathrm{eff}}$ the effective strain rate (considered as the second invariant of the strain-rate tensor). $A$
is assumed to be dependent on the temperature $T^*$ (temperature relative to the pressure melting point $T_{\mathrm{pmp}}$) and liquid water fraction $\omega$:

$$A = A(T^*, \omega) = \begin{cases} A_0\,e^{-Q_a/RT^*} & T^* < T_{\mathrm{pmp}} \\ A_0^t\,(1 + 181.25\omega) & T^* = T_{\mathrm{pmp}} \end{cases}, \tag{19}$$

where $A_0$ and $A_0^t$ are constants, $Q_a$ is the activation energy for creep, and $R$ is the gas constant. The constant $A_0^t$ is equal to $A(T^* = T_{\mathrm{pmp}}, \omega = 0)$. The upper bound of the water fraction $\omega$ is 0.01 to ensure validity of the flow rate factor parameterization
in the temperate part with the experimental dataset (Duval, 1977; Lliboutry and Duval, 1985).





For the dynamical model, we employ the higher-order Blatter-Pattyn approximation (Blatter, 1995; Pattyn, 2003). Basal sliding is allowed everywhere and the basal drag, $\tau_{\mathrm{b}}$, is written as:

$$\tau_{\mathrm{b},i} = -k^2 N v_{\mathrm{b},i}, \tag{20}$$

where $v_{\mathrm{b},i}$ is the basal velocity component in the horizontal plane and $i = x, y$ and $k^2$ the friction coefficient. The effective

pressure is defined as $N = \varrho_i \, g \, h$. At the ice front a zero pressure boundary condition is applied as all the ice is above sea level. A traction-free boundary condition is imposed at the ice/air interface.

For the thermal model, we impose a Dirichlet condition at the surface:

$$T(x,y) = 238.15\,\mathrm{K} + 1.67 \times 10^{-5}\,\mathrm{K\,m^{-1}} r. \tag{21}$$

The ice sheet base is subject to the decision chart presented in Aschwanden et al. (2012). In this implementation, the basal

boundary condition is allowed to switch between Neumann and Dirichlet type depending on the thermal basal conditions. The geothermal flux, $q_{\mathrm{geo}}$, is considered spatially constant.

To investigate the sensitivity of over- and under-stabilization, we perform experiments with three different stabilization formulations (Tab. 2). The setup SUPG maxK is the standard SUPG setup based on the maximum edge length of an element $K$ for the local mesh parameter $h_K$. In contrast, the SUPG minK uses the minimum edge length as, however, recommended

for anisotropic 2D meshes (Harari and Hughes, 1992; Mittal, 2000). Finally, the ASUPG is employed.

To study whether the stabilization is dependent on different mesh resolutions and the amount of advection, we vary the horizontal grid size and the amount of sliding. Here, we use a base mesh of 20 km in the interior, which is subsequently refined to $l_{\min} = (10, 5, 1)\,\mathrm{km}$ towards the glacier margin. The friction coefficient is treated as spatially constant and several experiments are performed with $k^2 = (400, 100, 50)\,\mathrm{a\,m^{-1}}$. For the three sliding cases, this results in frontal velocities of about

50, 350 and 1100 $\mathrm{m\,a^{-1}}$, respectively. We use 15 layers refined close to the base to account for the high velocity gradients and vertical shearing near the base in the vertical direction. The simulations are run 2000 years forward in time without the necessarily reaching a steady state.

## 4  Results and Discussion

### 4.1  Polythermal slab

The final steady-state CTS positions for all simulations are shown in Fig. 1. For the maximum value of temperate ice conductivity ($K_0/K_c = 10^{-1}$), the models result in a CTS position around 36–39 m. With decreasing $K_0/K_c$, the temperate ice layer thickness consistently decreases for the harmonic and geometric mean models and is almost halved for the lowest conductivity ratio $K_0/K_c = 10^{-5}$; the solution converges to the analytical CTS position for the high mesh resolution. However, for the harmonic mean, we detect a larger spread over the grid-resolutions at low $K_0/K_c$ compared to the geometric mean. The sim-

ulations with the arithmetic mean yield a completely different picture. The range in the CTS position increases considerably with decreasing $K_0/K_c$ and the analytical CTS position is met for the highest mesh resolution, below 2 m.

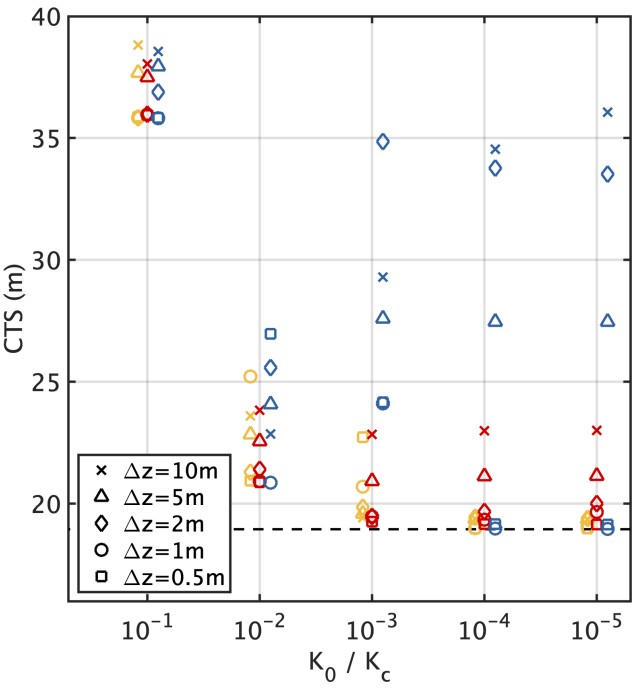

**Figure 1.** Comparison of simulated steady state CTS positions for different values of the temperate ice conductivity, $K_0$, for the polythermal slab experiment. The different conductivity models are shown as: arithmetic mean (blue), harmonic mean (red) and geometric mean (yellow). Results of different models are slightly shifted on the $x$-axis to not overlay each other. The dashed black line indicates the CTS position of the analytical solution derived for $K_0 = 0\,\mathrm{kg\,m^{-1}\,s^{-1}}$.

The steady-state results of the three conductivity models are verified with the analytical solution of the vertical enthalpy profile. Figure 2 shows the simulated vertical enthalpy profiles for $\Delta z = 10$ and $0.5\,\mathrm{m}$ and the lowest conductivity ratio $K_0/K_c = 10^{-5}$. The results of all models agree well with the analytical solution for high resolutions. At coarser resolu-

tions however, the simulated enthalpy profiles differ noticeably from the analytical solution for the arithmetic and the harmonic mean, while the geometric mean coincides well with the analytical solution. Please note, that the results for the harmonic mean are similar to those presented in Kleiner et al. (2015) for ISSM.

The accuracy of the simulations with the lowest conductivity ratio is measured with the root-mean-square error (RMSE) to the analytical solution. The RMSE as a function of vertical resolution is shown in Fig. 3. All three models exhibit different

behaviors. The arithmetic mean reveals a somewhat inconsistent behavior, while the harmonic mean shows approximately first-order convergence as $\Delta z \to 0$. Overall, the geometric mean shows low errors, and the error remains on a similarly low level even for coarse resolutions.

The different behaviors highlight the dependency of the solution on the CTS implementation details. As already identified by Kleiner et al. (2015) the usage of an arithmetic mean leads to oscillations in the enthalpy solution that are visible e.g. in



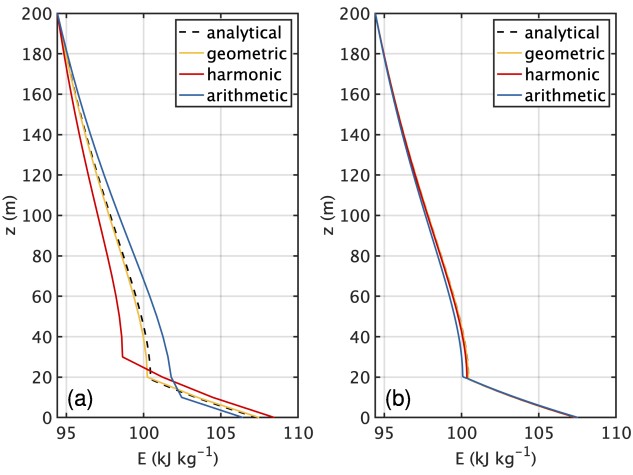

**Figure 2.** Simulated steady-state profiles of the enthalpy $E$ computed with the three conductivity models with $K_0/K_c = 10^{-5}$ and a vertical resolution of $\Delta z = 10\,\mathrm{m}$ (a) and $\Delta z = 0.5\,\mathrm{m}$ (b) compared to the analytical profile.

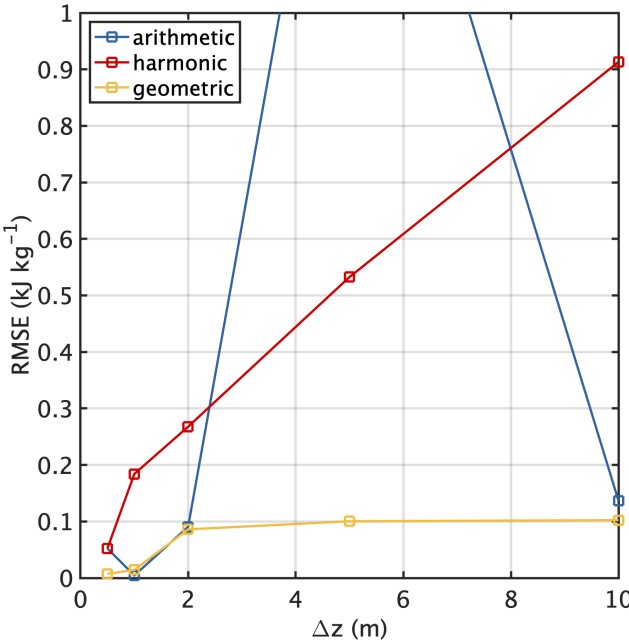

**Figure 3.** Root-mean-square error (RMSE) for the polythermal slab experiment. The RMSE is computed between the modeled enthalpy result and the analytical solution for different vertical grid resolutions $\Delta z$ and for each conductivity parameterization. Model results for arithmetic mean (blue), harmonic mean (red) and geometric mean (black) are obtained for the lowest conductivity ratio $K_0/K_c = 10^{-5}$.

a time-varying CTS position. Consequently, no steady-state solution is reached under these conditions. Here, wen applying a steady-state solver, the solver does not converge and the CTS position is flipping between the non-linear iterations.



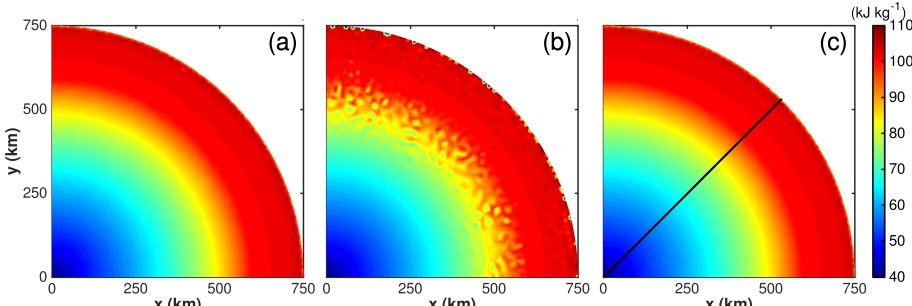

**Figure 4.** Simulated enthalpy (kJ kg$^{-1}$) for the ice dome experiment with $l_{\min} = 10\,\mathrm{km}$ and $k^2 = 50\,\mathrm{a\,m}^{-1}$. (a) SUPG maxK, (b) SUPG minK, (c) ASUPG. Black line in (c) indicates the location of the vertical profile shown in Fig. 5.

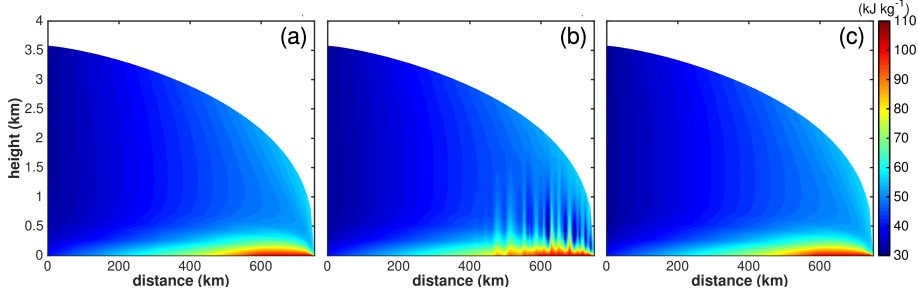

**Figure 5.** Simulated enthalpy (kJ kg$^{-1}$) for the ice dome experiment with $l_{\min} = 10\,\mathrm{km}$ and $k^2 = 50\,\mathrm{a\,m}^{-1}$ along a vertical cross section. (a) SUPG maxK, (b) SUPG minK, (c) ASUPG. The location of the vertical profile is shown in Fig. 4c.

### 4.2 Ice dome

In this experiment, we explore the impact of the parameter choices in the SUPG formulation on the reliability and accuracy of the results. In Fig. 4 the simulated basal enthalpy field is shown for the lowest resolution $l_{\min} = 10\,\mathrm{km}$ and high sliding case $k^2 = 50\,\mathrm{a\,m}^{-1}$ for the three employed stabilization formulations. Due to symmetry reason, only the upper-right part of the domain is shown. As expected, the SUPG minK produces unphysical oscillations in the simulated enthalpy field. SUPG maxK and ASUPG reveal a smooth result with merely minor oscillations a the ice front, where the surface slopes becomes singular. The same picture is observed along a vertical profile of the ice sheet interior (Fig. 5). For the SUPG minK, the numerical oscillations in the enthalpy field are visible in the whole ice column. The same qualitative behavior among the SUPG formulations is detected for all employed grid resolutions and sliding cases (Fig. 6). Increasing the mesh resolution leads to a significant reduction in upstream oscillations. However, oscillations still occur close to the ice margin. This is in line with the theory that $\tau_k$ must vanish as grid refinement increases, and no stabilization may be necessary for sufficiently fine meshes. The amount of basal sliding, which controls the amount of advection, plays a secondary role.

Surprisingly, SUPG maxK and ASUPG are visually indistinguishable and result in qualitatively similar results. However, when re-running the polythermal slab experiment with the three SUPG formulations, distinct differences in the sim-



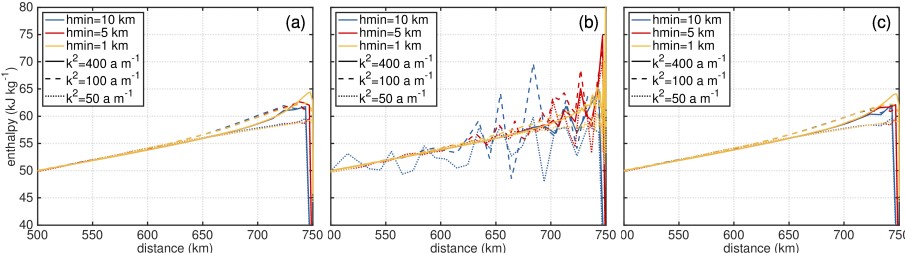

**Figure 6.** Simulated depth-averaged enthalpy (kJ kg$^{-1}$) for the ice dome experiment along a vertical cross section. (a) SUPG maxK, (b) SUPG minK, (c) ASUPG. The location of the vertical profile is shown in Fig. 4c.

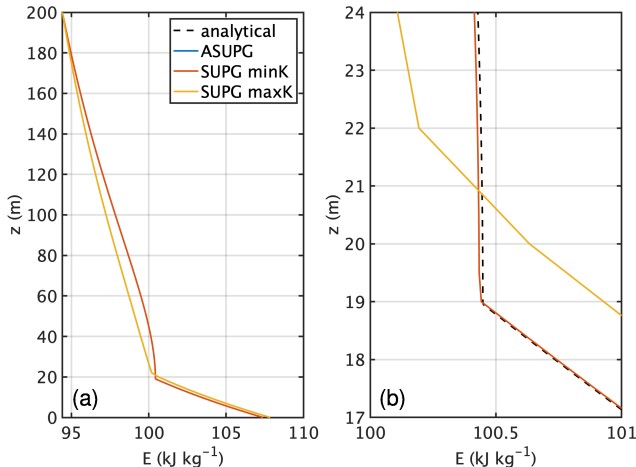

**Figure 7.** Simulated steady-state profiles of the enthalpy $E$ for the three different SUPG models by employing the geometric mean (Eq. 15) and a vertical resolution of $\Delta z = 0.5\,\mathrm{m}$ (a). Zoom to CTS region (b). Please note that ASUPG and SUPG minK overlay each other.

ulated enthalpy are obtained (Fig. 7). The simulations with ASUPG and SUPG minK both match the analytical solution with RMSE=0.01 and 0.01 kJ kg$^{-1}$, respectively. The simulation with SUPG maxK deviates considerably from the analytical solution with RMSE=0.48 kJ kg$^{-1}$. Overall, we find that (1) using SUPG maxK as the local mesh parameter results in an oscillation-free enthalpy field but tends to produce too much diffusion, (2) using SUPG minK as the local mesh parameter results in unphysically large oscillations for more complex geometries, and (3) ASUPG provided realistic solutions in all conducted experiments.

Our results demonstrate that choosing the stabilization parameter in a heuristic or ad-hoc manner, without knowledge of the possible effects, can impact the solution significantly. Choosing a sub-optimal value for the stabilization parameter can affect the accuracy of the solution, and result in over- or under-stabilization. The viability of the SUPG formulation strongly depends on appropriate parameter choices and in a worst-case scenario, the oscillations could cause the temperature to diverge. However, we have not investigated how the solution differences propagate to other components of an ice sheet model, e.g., by coupling to the evolution of the ice thickness.





Since the above-presented solutions for the ASUPG method are excellent, the parameter choices for the local mesh parameters $h_K^{\text{horizontal}}$, $h_K^{\text{vertical}}$, and the velocity norm $|\boldsymbol{v}|$ are not further investigated. The velocity norm is here treated equally in

both directions (Eq. 7), and no differentiation is made between the horizontal and vertical direction. Some test runs (not shown here) applying direction-dependent euclidean norms of the velocity revealed no discernible differences to the above-presented results. Additionally, in the current implementation, the local mesh parameter in the horizontal direction, $h_K^{\text{horizontal}}$, does not cover anisotropy of elements in the horizontal plane. However, these simplifications have so far not led to numerical problems, but might be subject to future work.

## 275   5   Conclusions

We presented extended enthalpy formulations within the ice flow model ISSM compared to Seroussi et al. (2013) and Kleiner et al. (2015). Treating the discontinuous conductivity as a geometric mean results in a good solution for coarse resolutions compared to the analytical solution. This treatment is an improvement compared to earlier ISSM results presented in Kleiner et al. (2015) and based on a harmonic mean.

Additionally, we tested various SUPG stabilization formulations on their ability to deal with the high aspect ratio of 3D elements in glaciological applications. We found that the traditional parameters in the SUPG stabilization coefficients are susceptible to stabilization parameter choices, here the local mesh parameter which is easily adjustable. We propose a novel anisotropic SUPG (ASUPG) method that circumvents the high aspect-ratio problem in ice sheet modelling by treating the horizontal and vertical direction separately in the stabilization coefficients. The ASUPG method provides accurate results for

the thermodynamic equation on geometries with very small aspect ratios like ice sheets.

*Code availability.* The ice flow model ISSM (Larour et al., 2012) is open source and freely available at https://issm.jpl.nasa.gov/ (last access: March 20, 2020). ISSM version 4.17 is currently under development but will be released to the public in the very close future.

*Author contributions.* MR conducted the study supported by the other authors. MR set up the experiments conducted with ISSM and analysed the experiments. MM and HS provided technical ISSM support. MR wrote the manuscript together with the other authors.

*Competing interests.* The authors declare that they have no conflict of interest.

*Acknowledgements.* Martin Rückamp acknowledges support of the Helmholtz Climate Initiative REKLIM (Regional Climate Change). For discussions and suggestions we thank Vadym Aizinger (University Bayreuth), Yonqi Wang (University Darmstadt) and Luca Wester (University of Erlangen).





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





**Table 1.** Used constants and model parameters.

| Quantity | Value | Unit |
|---|---|---|
| Seconds per year, spy | 31556926 | $\mathrm{s\,a^{-1}}$ |
| Gravitational acceleration, $g$ | 9.81 | $\mathrm{m\,s^{-2}}$ |
| Density of ice, $\varrho_i$ | 910 | $\mathrm{kg\,m^{-3}}$ |
| Density of water, $\varrho_w$ | 1000 | $\mathrm{kg\,m^{-3}}$ |
| Reference temperature, $T_{\mathrm{ref}}$ | 223.15 | K |
| Melting point at standard pressure, $T_0$ | 273.15 | K |
| Specific heat capacity, $c_{\mathrm{i}}$ | 2009.0 | $\mathrm{J\,kg^{-1}K^{-1}}$ |
| Thermal conductivity, $k_{\mathrm{i}}$ | 2.1 | $\mathrm{W\,m^{-1}K^{-1}}$ |
| Glen exponent, $n$ | 3 | |
| **Polythermal slab:** [a] | | |
| Ice thickness, $H$ | 200 | m |
| Geothermal heat flux, $q_{\mathrm{geo}}$ | 0.0 | $\mathrm{W\,m^{-2}}$ |
| Latent heat of fusion, $L$ | $3.35 \times 10^5$ | $\mathrm{J\,kg^{-1}}$ |
| Clausius-Clapeyron constant, $\beta$ | 0.0 | $\mathrm{K\,Pa^{-1}}$ |
| Rate-factor, $A$ | $5.3 \times 10^{-24}$ | $\mathrm{Pa^{-3}s^{-1}}$ |
| Temperate ice conductivity, $K_0$ | $k_{\mathrm{i}}/c_{\mathrm{i}} \times 10^{-1}$ $\vdots$ $k_{\mathrm{i}}/c_{\mathrm{i}} \times 10^{-5}$ | $\mathrm{kg\,m^{-1}s^{-1}}$ |
| **Ice dome:** [b] | | |
| Maximum thickness, $h_{\mathrm{max}}$ | 3575.1 | m |
| Maximum extent, $r_{\mathrm{max}}$ | 750 | km |
| Geothermal heat flux, $q_{\mathrm{geo}}$ | 0.042 | $\mathrm{W\,m^{-2}}$ |
| Latent heat of fusion, $L$ | $3.34 \times 10^5$ | $\mathrm{J\,kg^{-1}}$ |
| Clausius-Clapeyron constant, $\beta$ | $9.8 \times 10^{-8}$ | $\mathrm{K\,Pa^{-1}}$ |
| Temperate ice conductivity, $K_0$ | $k_{\mathrm{i}}/c_{\mathrm{i}} \times 10^{-2}$ | $\mathrm{kg\,m^{-1}s^{-1}}$ |
| Universal gas constant, $R$ | 8.314 | $\mathrm{J\,mol^{-1}K^{-1}}$ |
| Activation energy for creep, $Q_a$ | $6 \times 10^4$ if $T^* < 263.15\,\mathrm{K}$ | $\mathrm{kJ\,mol^{-1}}$ |
| | $13.9 \times 10^4$ if $T^* > 263.15\,\mathrm{K}$ | $\mathrm{kJ\,mol^{-1}}$ |
| Constant of proportionality, $A_0$ | $3.61 \times 10^{-13}$ if $T^* < 263.15\,\mathrm{K}$ | $\mathrm{Pa^{-3}s^{-1}}$ |
| | $1.73 \times 10^3$ if $T^* > 263.15\,\mathrm{K}$ | $\mathrm{Pa^{-3}s^{-1}}$ |

[a] based on Greve and Blatter (2009)

[b] based on Vialov (1958) and Payne et al. (2000)





**Table 2.** List of employed stabilization approaches

| experiment label | description |
| --- | --- |
| SUPG maxK | SUPG formulation (Eq. 6) with $h_k$ as the maximum edge of the 3D element $K$ |
| SUPG minK | SUPG formulation (Eq. 6) with $h_k$ as the minimum edge of the 3D element $K$ |
| ASUPG | anisotropic SUPG (Eqs. 10, 11 and 12) formulation with $h_K^{\mathrm{horizontal}}$ and $h_K^{\mathrm{vertical}}$ |