# Peer review of "Extended enthalpy formulations in the ice flow model ISSM version 4.17: discontinuous conductivity and anisotropic SUPG"

_Geoscientific Model Development, 2020_

## Referee Comment (RC1) · Stephen Cornford (Referee) · 22 May 2020

**1   General comments**

This paper introduces two improvements to the thermodynamics represented in the ISSM (Ice Sheet System Model). ISSM is one of most widely used and advanced large scale ice sheet models in the world, and correct treatment of the thermodynamics is needed for many applications. The improvements are related to the discretization scheme rather than physics, but are analyzed in the context of the ice sheet physics. They are well enough described for other developers to see how to implement them,

and examples are given that show their benefits. Given that, I think this is a suitable paper for inclusion in GMD.

I do think that some minor attention to the manuscript is in order.

**2 Specific comments**

1. L17-20 - perhaps include some examples, e.g the thermomechanical instability discussed by e.g Hindmarsh 2009.

2. L35 'Numerical instabilities inherent to the advection component of this equation tend to occur without stabilizing the standard Galerkin finite element method.' and indeed, any other method.

3. L82 'The temperate ice conductivity'. Expand on this a bit: say what it means physically (e.g transport of latent heat down a moisture gradient and often against a temperature gradient), and indicate typical literature value for $K_0/K_c$ (including zero)

4. L130. Are $S_1$ and $S_2$ introduced just fit the equations on the page? That is the impression I get. But I wondered if the sources cited also split $S$ this way and take some specific interest in each term.

5. L140 (and after) How is $\theta$ found? And is $\theta$ a volume?

6. 149 'The applicability of the three models is controversial in the literature and depends strongly on the problem' - citation/examples would help here.

7. 'Since heat conduction through porous media is likely a combination of both structures, a geometric mean can be interpreted as accounting for both processes as

it always results in a value in between an arithmetic and harmonic mean' - but so would a number of combinations, and you might imagine trying to weight them.

8. Figure 1. It is hard to make out the order of the symbols especially for the geometric mean (because they are close together) One solution could be to plot $|CTS(\Delta z, K_0)$ - $CTS(K_0 = 0, \Delta z = 0)|$ (|numerical solution - analytic solution|) on a log scale, though that might only help with the smaller $K_0$ cases. A sharp eye might then tell what the rate of convergence was, both as $K_0 \to 0$ and $\Delta z \to 0$.

9. Figure 3. log scales for both $x-$ and $y-$ axes would help to make this figure clear, perhaps with indicative rates $(\Delta z)^n$ for suitable $n$ I would also plot $1/\Delta z$ on the $x-$axis rather than $\Delta z$ (so left -> right has increasing number of DoF but that really is a minor detail)

10. Figure 7 the caption does state that ASUPG and SUPG minK overlay one another, but if one line was dashed (or thicker) that could be apparent in the figure.

**3 Technical corrections / very minor copy editing suggestions**

1. Abstract, first line: 'ice sheets' should be 'ice sheet'?

2. L30 An increasing number of ice flow models is adopting → ...are adopting, or ...have adopted

3. L38 convection-dominated. You used 'advection' in a previous line so I would stick with it.

4. 'The aspect ratio of anisotropic grid-cells in the FEM is particularly problematic' → 'Low aspect ratio mesh elements in the FEM are particularly problematic'?

5. ISMs are dealing with very thin geometries - maybe say 'low aspect ratio' here to be clear.

6. 'For instance, a . . . ' → 'A . . . '

7. 'Our work is indeed inspired by the' → 'Our work addresses'

8. L83 'At the upper surface, Dirichlet boundary conditions are imposed'. In this case - but potentially a heat flux might be imposed if coupled with a snow pack model.

9. L100 'bilinear elements' → Piecewise linear?

10. L111 'Once the elements become anisotropic or distorted'. Is the 'distorted' help-ful here?

11. L159 'We run' or 'We ran' - not so important I guess but 'methods in the past, results in the present'

12. L165 'The setup poses a *reasonable* situation in glacier modelling' typical? rep-resentative?

13. L167 'The horizontal velocity...' (and does not vary horizontally)

14. 174 In this set-up, no stabilization is applied, i.e. the term $S(E, w)$ in Eq. 4 is ignored. (Because Pe is small I suppose? but is that the case below the CTS?)

15. L214 'without *the* necessarily reaching a steady state' (remove the)

16. L221 (and onward) 'CTS position'. CTS elevation?

17. L245 'Due to symmetry reason, only' → 'Due to symmetry, only'

18. 'too much diffusion' → 'far more diffusion than the other choices?'

19. L265 'the oscillations could cause the temperature to diverge'. From what? In one sense they do already (from the solution, as $h_K$ grows), but I think you mean numerical error so severe that it becomes grossly unphysical (e.g $E < 0$, or $E$ very large) and/or numerical error so severe it causes an iterative solver to produce successively worse approximations (blow up).

20. L277 'Treating the discontinuous conductivity as a geometric mean'. A bit of rephrasing is needed: the conductivity is not treated as any kind of mean, rather, a particular formula is used when estimating $K_{\text{eff}}$ at various points.

---

## Referee Comment (RC2) · Alexander Robinson (Referee) · 18 Jun 2020

This manuscript describes the formulation of the thermodynamics-enthalpy solver in the ice-sheet model ISSM, including a novel numerical treatment of the discontinuous boundary between temperate and frozen ice within an ice column (known as the CTS). This problem is very important and relevant for ice-sheet modeling today, as more models begin to make use of the enthalpy formulation to simulate polythermal ice. I find the manuscript well written and clear. The methods, including new parameterizations applied here, and the results are straightforward to understand. The benchmark tests make the problem clear, and show the impact of the solution proposed by the authors.

I therefore suggest that the manuscript be published with only very minor technical revisions.

Minor copyediting comments:

P1L10: are not accounting for => do not account for

P10L240: wen => when

P10L241: is flipping => flips

P11L245: symmetry reason => symmetry

P11L249: whole ice column => whole ice profile [I don't see oscillations within a given column]

P13L271: euclidean => Euclidean

Figures: consider using a different/darker color than yellow for the geometric mean points/curves. Since this is the novel result, it would be valuable for it to stand out a bit more in the figures.

---

## Author Comment (AC1) · 17 Jul 2020

We would like to thank the reviewers for their constructive comments that helped to improve the manuscript 'Extended enthalpy formulations in the ice flow model ISSM version 4.17: discontinuous conductivity and anisotropic SUPG'. We have revised the manuscript accordingly and will be happy to provide a new manuscript.

Please find below the reviewer's comments in black and a point-by-point response in blue.

**Review #1**

**1 General comments**

This paper introduces two improvements to the thermodynamics represented in the ISSM (Ice Sheet System Model). ISSM is one of most widely used and advanced largescale ice sheet models in the world, and correct treatment of the thermodynamics is needed for many applications. The improvements are related to the discretization scheme rather than physics, but are analyzed in the context of the ice sheet physics. They are well enough described for other developers to see how to implement them, and examples are given that show their benefits. Given that, I think this is a suitable paper for inclusion in GMD. I do think that some minor attention to the manuscript is in order.

We would like to thank Stephen Cornford for the positive feedback.

**2 Specific comments**

1. L17-20 - perhaps include some examples, e.g the thermomechanical instability discussed by e.g Hindmarsh 2009.
Done. We included MacAyeal (1993), Hindmarsh 2009 and Feldmann and Leermann (2017) as prominent examples.

2. L35 'Numerical instabilities inherent to the advection component of this equation tend to occur without stabilizing the standard Galerkin finite element method.' And indeed, any other method.
The paragraph is rewritten to: "*In ISMs, the governing thermodynamic equation are discretized, e.g. using the finite element method (FEM). Special care has to be taken to the parabolic thermodynamic equation as numerical instabilities inherent to the advection component of this equation tend to occur without stabilization. When employing the FEM the standard Galerkin finite element method is often stabilized with the popular Streamline Upwind Petrov–Galerkin (SUPG) method (Brooks_Hughes, 1982).*"

3. L82 'The temperate ice conductivity'. Expand on this a bit: say what it means physically (e.g transport of latent heat down a moisture gradient and often against a temperature gradient), and indicate typical literature value for $K_0/K_c$ (including zero)
We expand and rephrased this section. See beginning of chapter 2.1 in the new version of the manuscript or in the marked-up version attached to this response.

4. L130. Are $S_1$ and $S_2$ introduced just fit the equations on the page? That is the impression I get. But I wondered if the sources cited also split S this way and take some specific interest in each term.
$S_1$ and $S_2$ were just introduced for a clearer presentation. But you are right, there is no need to split the equation into $S_1$ and $S_1$. In the updated version we present $S^{ASUPG}$ in one single equation.

5. L140 (and after) How is θ found? And is θ a volume?

Θ is a volume fraction. We rephrased the sentences here and give an explanation how θ is calculated.

6. 149 'The applicability of the three models is controversial in the literature and depends strongly on the problem' - citation/examples would help here.

You are right, we added a list of references.

7. 'Since heat conduction through porous media is likely a combination of both structures, a geometric mean can be interpreted as accounting for both processes as it always results in a value in between an arithmetic and harmonic mean' - but so would a number of combinations, and you might imagine trying to weight them.

Yes, of course one could design a model as a combination of the arithmetic and harmonic mean that gives similar/comparable results. Indeed, we found studies were such combinations are proposed. We added a sentence and included the references.

8. Figure 1. It is hard to make out the order of the symbols especially for the geometric mean (because they are close together) One solution could be to plot $|CTS(\Delta z, K_0) - CTS(K_0 = 0, \Delta z = 0)|$ (|numerical solution - analytic solution|) on a log scale, though that might only help with the smaller $K_0$ cases. A sharp eye might then tell what the rate of convergence was, both as $K_0 \rightarrow 0$ and $\Delta z \rightarrow 0$.

Thanks, that's a good recommendation. We updated the figure accordingly.

9. Figure 3. log scales for both x− and y−axes would help to make this figure clear, perhaps with indicative rates $(\Delta z)^n$ for suitable n. I would also plot $1/\Delta z$ on the x−axis rather than $\Delta z$ (so left -> right has increasing number of DoF but that really is a minor detail)

Again, we updated the figure accordingly.

10. Figure 7 the caption does state that ASUPG and SUPG minK overlay one an-other, but if one line was dashed (or thicker) that could be apparent in the figure.

In the updated figure, we draw the SUPG minK with a thicker line.

**3 Technical corrections / very minor copy editing suggestions**

1. Abstract, first line: 'ice sheets' should be 'ice sheet'?

Done.

2. L30 An increasing number of ice flow models is adopting → ...are adopting, or...have adopted

We changed to "… are adopting ...".

3. L38 convection-dominated. You used 'advection' in a previous line so I would stick with it.

We sticked to "advection".

4. 'The aspect ratio of anisotropic grid-cells in the FEM is particularly problematic'→'Low aspect ratio mesh elements in the FEM are particularly problematic'?

Changed as suggested.

5. ISMs are dealing with very thin geometries - maybe say 'low aspect ratio' here to be clear.

Changed as suggested.

6. 'For instance, a . . . '→'A . . . '
Done.

7. 'Our work is indeed inspired by the'→'Our work addresses'
Done.

8. L83 'At the upper surface, Dirichlet boundary conditions are imposed'. In this case - but potentially a heat flux might be imposed if coupled with a snow packmodel.
Well, to our knowledge most of the ISM application are not making use of a firnmodel. Most of the ISMs prescribing temperature/climate data from GCM/RCM products, which are imposed as Dirichlet. We think this is detail that should not be mentioned here. However, we have rewritten the sentence to: "Dirichlet boundary conditions are imposed at the upper surface in all setups."

9. L100 'bilinear elements'→Piecewise linear?
Changed as suggested.

10. L111 'Once the elements become anisotropic or distorted'. Is the 'distorted' helpful here?
We dropped "distorted".

11. L159 'We run' or 'We ran' - not so important I guess but 'methods in the past, results in the present'
Done.

12. L165 'The setup poses a *reasonable* situation in glacier modelling' typical? representative?
Yes, maybe 'representative' applies better.

13. L167 'The horizontal velocity...' (and does not vary horizontally)
Done.

14. 174 In this set-up, no stabilization is applied, i.e. the term S(E,w) in Eq. 4 is ignored. (Because Pe is small I suppose? but is that the case below the CTS?)
We applied no stabilization in order to be comparable to the ISSM results already published in Kleiner et al. (2015). The Pe number would require some stabilization in particular for small Keff values (below the CTS). However, adding some consistent stabilization (SUPG or ASUPG) does not alter the results drastically. We clarified that in the updated version of the manuscript.

15. L214 'without *the* necessarily reaching a steady state' (remove the)
Done.

16. L221 (and onward) 'CTS position'. CTS elevation?
Done. Changed in the whole manuscript.

17. L245 'Due to symmetry reason, only'→'Due to symmetry, only'
Done.

18. 'too much diffusion'→'far more diffusion than the other choices?'
Changed as suggested.

19. L265 'the oscillations could cause the temperature to diverge'. From what? In one sense they do already (from the solution, ash $\kappa$ grows), but I think you mean numerical error so severe that it becomes grossly unphysical (e.g E <0, or E very large) and/or numerical error so severe it causes an iterative solver to produce successively worse approximations (blow up).

We have rephrased it to: "…the oscillations could cause unphysical values or the solver to diverge."

20. L277 'Treating the discontinuous conductivity as a geometric mean'. A bit of rephrasing is needed: the conductivity is not treated as any kind of mean, rather, a particular formula is used when estimating $K_{eff}$ at various points.

We added 'Treating … at the CTS as geometric mean'.

**Review #2**

This manuscript describes the formulation of the thermodynamics-enthalpy solver in the ice-sheet model ISSM, including a novel numerical treatment of the discontinuous boundary between temperate and frozen ice within an ice column (known as the CTS). This problem is very important and relevant for ice-sheet modeling today, as more models begin to make use of the enthalpy formulation to simulate polythermal ice. I find the manuscript well written and clear. The methods, including new parameterizations applied here, and the results are straightforward to understand. The benchmark tests make the problem clear, and show the impact of the solution proposed by the authors. I therefore suggest that the manuscript be published with only very minor technical revisions.

We would like to thank Alex Robinson for his positive evaluation.

Minor copy editing comments:

P1L10: are not accounting for => do not account for
Done.

P10L240: wen => when
Done.

P10L241: is flipping => flips
Done.

P11L245: symmetry reason => symmetry
Done.

P11L249: whole ice column => whole ice profile [I don't see oscillations within a given column]
Yes, you are right.

P13L271: euclidean => Euclidean
Done.

Figures: consider using a different/darker color than yellow for the geometric mean points/curves. Since this is the novel result, it would be valuable for it to stand out a bit more in the figures.
We changed the colors: the geometric mean appears a red points/line in the updated figures.

[revised manuscript text omitted]